# Migraine Treatment Using Erenumab: Can Lead to a Cognitive and Psychological Qualitative Improvement?

**DOI:** 10.3390/medicina59050936

**Published:** 2023-05-12

**Authors:** Michele Torrisi, Francesco Corallo, Viviana Lo Buono, Marcella Di Cara, Rosario Grugno, Riccardo Lo Presti, Angelo Quartarone, Maria Cristina De Cola

**Affiliations:** IRCCS Centro Neurolesi “Bonino-Pulejo” S.S. 113 Via Palermo, C/da Casazza, 98124 Messina, Italyriccardo.lopresti@irccsme.it (R.L.P.);

**Keywords:** cognitive functions, erenumab, migraine

## Abstract

Migraine is one of the most disabling disorders in the world, associated with poor quality of life. Migraine prevention strategies have increasingly evolved since monoclonal antibodies against the calcitonin gene-related peptide (CGRP), or its receptor, were identified. CGRP is the ideal target of monoclonal antibodies (mAbs). In particular, erenumab is the mAb that has shown good therapeutic efficacy in reducing pain intensity and having high tolerability. In this study, we aimed to investigate the efficacy of erenumab on both cognitive performance and psychological well-being. This was a pilot study with a retrospective design that included 14 subjects (2 males and 12 females), with a mean age of 52.29 ± 9.62, who attended the Headache and Migraine outpatient clinic of the IRCCS Centro Neurolesi Bonino-Pulejo of Messina. The evaluation consisted of measuring cognitive and psychological functioning. Comparing clinical and psychometric test scores between baseline and follow-up, we found a significant improvement in both cognitive performance and quality of life. We also observed a decrease in migraine disability. Our findings have shown improvements in global cognitive performance and quality of life in migraine patients taking erenumab.

## 1. Introduction

Migraine is a neurological disorder with a complex physiopathology affecting around 15% of the global population. It is more common in women relative to men and is the second most disabling disease in the world [1,2]. Migraine is believed to be due to a mixture of environmental and genetic factors. Research on the neurobiology of migraine has evolved over time and has included various hypotheses and theories on both underlying physiopathology and therapeutic development [3]. Despite this, migraine remains a complex and symptomatically heterogeneous disorder with a significant therapeutic gap in terms of drugs considered effective and well-tolerated for its treatment [4]. The updated International Classification of Headache Disorders, 3rd Edition [5] describes the phenotype of migraine in an increasingly complex and enriched way. It has been shown that many migraine patients experience symptoms of prodromal before the onset of the actual episode, symptoms that may begin hours or days before the pain [6]. Three phases have been described in the classic migraine attack: the prodrome phase (premonitory phase), the headache phase, and the postdrome phase. Not all of these stages are experienced by each patient or during each attack. The premonitory phase can occur from hours to days before an attack and is often characterized by symptoms such as fatigue, impaired concentration, neck stiffness, sensitivity to light and noise, nausea, and food cravings. In the second phase, there is frontal headache, unilateral and pulsating of varying intensity, often accompanied by nausea, vomiting, blurred vision, photophobia, and phonophobia. The postdrome phase is instead characterized by symptoms that can persist until a few days after the onset of the attack such as fatigue, concentration disorders, neck stiffness, and sensitivity to light and noise.

Typically, migraine is characterized by typically unilateral, often severe, pain throbbing with associated features such as hypersensitivity to multiple stimuli, including visual (photophobia), auditory (phonophobia), and sensory (cutaneous allodynia) stimuli, during migraine attacks. Migraine is a major cause of disability [7]. Because of the severe degree of debilitation due to migraine, sufferers have significant limitations in daily life with effects on emotional-behavioral and relational aspects [8,9]. Migraine indeed impacts the subjective well-being and the quality of life, causing missed days of school or work, reducing productivity, and producing financial burden with the cost of the medications [10]. The frequent migraine attacks can cause functional impairments, involving both physical and psychological effects.

To date, available treatments for migraine have included drugs not specifically developed for the treatment of this disease, such as beta-blockers, antiepileptic drugs, selective serotonin reuptake inhibitors (SSRIs), and injectable therapies such as botulinum toxin, approved for the treatment of other diseases and subsequently administered for the therapeutic management of migraine. 

These drugs have been used alone or in combination, with varying efficacy and tolerability [11]. Despite advances in the management of headache disorders, some patients with migraine do not experience adequate pain relief with acute and preventive treatments. The terms “refractory” and “intractable” headache have been used to describe this particular condition.

In addition, decreased compliance is very common: it is estimated that less than 25% of patients adhere to the first prescribed therapy after one year, due to side effects or poor efficacy [12,13]. This often leads migraine patients to medication overuse headache (MOH), a condition characterized by overuse of different acute medications that can paradoxically worsen headache, disability, and quality of life [14]. In recent years, treatment of migraine has seen the development of monoclonal antibodies that specifically target calcitonin gene-related peptide (CGRP): eptinezumab, fremanezumab, and galcanezumab, which bind to the CGRP molecule, and erenumab that binds CGRP receptors [15]. This treatment represents an extension of the therapeutic options, which already exist in migraine prevention. In randomized, placebo-controlled studies, the efficacy and good tolerability of these specific substances have been demonstrated in patients with episodic and chronic migraine [16]. The main standard adopted for prescribing these drugs is the presence of headache resistance to treatment with at least three preventative classes of antimigraine drugs, as clinical guidelines and the European Medicines Agency have established according to published evidence [17].

Erenumab is a fully human antibody and is the only mAb acting on the CGRP pathway by blocking its receptor. It is the first in the CGRP mAb class approved by the US Food and Drug Administration (May 2018) and the European Medicines Agency (July 2018). The recommended dosage of erenumab is 70 mg/mL injected subcutaneously once monthly, although some patients may need 140 mg/mL once monthly [18]. Current evidence demonstrated that erenumab is effective and well-tolerated in preventive therapy of migraine, both episodic and chronic [19,20].

As shown by the literature, migraine has a significant negative impact on psychological well-being, as well as causing a decline in multiple areas of sexuality and worsening marital satisfaction [21,22]. Anxiety and mood disorders have also been reported as the most prominent psychiatric comorbidities associated with migraine [23]. These psychological disturbances influence prognosis, response to treatments, and clinical outcomes, also depending on headache frequency and diagnostic category (e.g., migraine or tension-type headaches) [24,25,26]. In addition, several studies have found that migraine also affects cognitive functions [27]. Indeed, migraine patients showed poorer cognitive performance in all domains during interictal and ictal phases when compared to healthy control subjects [28].

In this pilot study, we aim to observe the effect of the specific monoclonal antibody “erenumab” on the cognitive performance and psychological well-being of a small number of migraine patients in order to obtain preliminary data for further studies concerning the multidimensional evaluation of this disorder.

## 2. Materials and Methods

### 2.1. Study Design and Population

This was a pilot study with a retrospective design. Subjects attended the Headache and Migraine outpatient clinic of the IRCCS Centro Neurolesi Bonino-Pulejo of Messina. The diagnosis of migraine was performed by two neurologists, specialists in headache disorders, according to International Headache Society criteria [29]. Patients eligible for the study suffered from chronic migraine for at least 10 years and failed at least three preventive treatments.

Exclusion criteria were: (1) other types of headaches; (2) vascular disease or trauma; (3) history of major psychiatric disorders; (4) presence of metabolic disorders; and (5) other neurological conditions. In fact, the presence of such pathologies would not allow a determination of the possible correlation between cognitive or mood disorders and migraine.

These outpatients underwent erenumab every 28 days, 70 mg/mL injected subcutaneously. The hospital protocol included written consent to the treatment and cognitive evaluations every 3 months. The outpatient procedure involves neuropsychological evaluation at the first drug administration (T0) and at the fourth drug administration (T1).

A total of 14 subjects (2 males and 12 females), with a mean age of 52.29 ± 9.62, met inclusion/exclusion criteria and were selected for this study. The characteristics of the sample are reported in Table 1.

Data were retrospectively collected from medical records. Although Ethics Committee approval was not necessary, in accordance with the current regulation of our hospital, all participants signed written informed consent at hospital admission to use their information for research purposes.

### 2.2. Assessment

We used the Montreal Cognitive Assessment-MOCA to evaluate the presence of cognitive alterations. The MoCA checks different types of cognitive or thinking abilities: attention and concentration, executive functions, memory, language, visuoconstructional skills, conceptual thinking, calculations, and orientation. To evaluate anxiety and depressive symptoms, the Beck Depression Inventory (BDI-II) and Hamilton Rating Scale for anxiety (HAM-A) were administered. The BDI-II is characterized by 21 items and measures the severity of depressive symptoms. For each item, there are four possible answers, with a score from 0 to 3. The maximum total score is 63, where 0–13 indicates minimal depression, 14–19 mild depression, 20–28 moderate depression, and >29 severe depression. Severity of anxiety symptoms is measured by the HAM-A. The scale consists of 14 items that measure psychic anxiety and somatic anxiety. Each item is scored on a scale of 0 (not present) to 4 (severe), with a total score range of 0–56, where <17 indicates mild severity, 18–24 mild to moderate severity, and 25–30 moderate to severe. The Migraine Disability Assessment (MIDAS) was used to assess the severity of disability during daily activities (work, home and family commitments, leisure, or social activities). The migraine disability was graded in four classes according to MIDAS scores: 0–5 as minimal, 8–10 as mild, 11–20 as moderate, and 21 or more as severe disability. Finally, the Short Form Health Survey 36 (SF-36) was used for the overall perceived quality of life. The SF-36 includes one multi-item scale that assesses eight health concepts: (1) limitations in physical activities because of health problems; (2) limitations in social activities because of physical or emotional problems; (3) limitations in usual role activities because of physical health problems; (4) bodily pain; (5) general mental health (psychological distress and well-being); (6) limitations in usual role activities because of emotional problems; (7) vitality (energy and fatigue); and (8) general health perceptions. These easy-to-use self-administered patient questionnaires are validated and widely used for clinical research.

### 2.3. Statistical Analysis

We performed a non-parametric analysis because of the reduced sample dimensionality and the non-normal distribution of most target variables detected by means of the Shapiro-Wilk test. Thus, we used the Wilcoxon signed-rank test to compare the assessment scores between T0 and T1. Effect sizes were assessed by dividing the test statistic by the square root of the number of observations. The Chi-squared test was used to compare proportions. Continuous variables were expressed in median ± first-third quartile, whereas categorical variables were in frequencies and percentages. Statistical analysis was performed by using the 4.0.5 version of the open-source software R. A *p* < 0.05 was considered as the significance level.

## 3. Results

Comparing the clinical and psychometric test scores between baseline and follow-up, we found significant improvement in MOCA scores (*p* < 0.01, d = −0.65), as well as in scores of SF-36 Physical functioning (*p* = 0.04, d = −0.44). In detail, in three out of eight subscales of the SF-36, the scores were significantly improved: Pain (*p* = 0.03, d = −0.50), General health (*p* = 0.03; d = −0.49), and Energy/fatigue (*p* < 0.01; d = −0.65), as shown in Table 2. We also observed a decrease in migraine disability, although pre-post MIDAS changes did not reach statistical significance.

## 4. Discussion

This pilot study describes the effect of erenumab on cognitive functioning and psychological well-being in a sample of migraine patients. These aspects have been poorly addressed in previous studies on therapy by CGRP, which have generally focused on the efficacy of physical symptoms [30].

Literature data reported discordant results about the migraine effect on cognitive functioning. Many migraineurs often complain of intellectual impairment, particularly deficits in attention and memory, but also confusion during sudden stabbing migraine probably attributable to physical symptoms, such as pain, nausea and photophobia, that decreases cognitive efficiency [31]. Most of the clinic-based studies reported worse cognitive performance of migraine patients on tests of verbal and visuospatial memory, information processing speed, executive function, and attention even during intercritical periods. Disease duration, frequency and duration of migraine attacks, and pain intensity may be factors in cognitive impairment in migraine patients. Huang et al. [32] showed that the increased frequency and the longer durations of migraine attacks are correlated with worse cognitive performance. The migraine management requires, as an acute or prophylactic line of intervention, the frequent use of drugs acting on the central nervous system. The long-term impact of these medications on cognition and neurodegeneration has never been consistently assessed despite the different anti-migraine drugs showing distinct profiles concerning the risk of cognitive deficits [33].

To date, there are no more data on the efficacy of erenumab on cognitive functioning. Our findings have shown improvements in global cognitive performance in migraine patients taking erenumab. In addition, we found that pain intensity significantly decreased in association with the numerical reduction in the number of monthly migraine days. The reduction in headache days, and consequently, less disability caused by the disease, resulted in a greater subjective well-being with an increase in the quality of life. Reduced pain intensity is an important outcome measure which is meaningful for patients, as it has an impact on migraine-related disability. Additionally, Lipton et al. [34] described that in the patients treated with erenumab, there were improvements in pain intensity scores regardless of the threshold used, with a clear separation observed between the placebo and erenumab in the cumulative distribution of change. The benefits of preventive treatment included reducing the number of monthly migraine days’ frequency and reduction in the maximum pain intensity of breakthrough. 

In contrast to other studies [19,35], we did not observe a benefit of monoclonal antibodies on depressive or anxiety states. To date, many studies reported an association between migraine and some mental impairment, underlining that more than 25% of migraineurs meet the criteria for mood alterations [36]. It is well-known, the comorbidity between migraine and mood disorders, that if left untreated, it can increase migraine-related disability, reduce quality of life, and negatively impact treatment outcomes [37]. In addition, psychiatric comorbidities can play a significant role in how one perceives pain, copes with it, and maintains a normal lifestyle. 

Migraine and symptoms of anxiety or depression seem to be bidirectionally linked, where the presence of one disorder increases the risk and severity of the other. Comorbid psychiatric conditions need to be considered when devising a treatment plan for the patient with migraine. The studies on the impact of psychiatric comorbidities on the effectiveness of migraine treatment are mixed. Some authors have cited psychiatric comorbidities as a reason for the failure of migraine treatment [38]. Other studies have indicated that patients with comorbid psychiatric diagnoses show similar rates of improvement after treatment when compared with patients with no comorbid psychiatric diagnoses. To date, the exact temporal relationship between depressive symptoms or mood changes and the various phases of the migraine attack has not yet been examined. 

In our opinion, depressive or anxious conditions, once established, take several months to weaken, but unfortunately, the COVID-19 pandemic led to changes in the management of outpatients, suspending routine psychological assessments. It has led to a reduction in visits and the lack of long-term evaluations, involving some limitations of the study, such as the small sample size. Indeed, it would be interesting to verify the effects of erenumab on a larger group of subjects. In addition, a longer observation period (e.g., 12 months) might bring out psychological changes.

Migraine is associated with a substantially greater personal and societal burden and higher frequency of comorbidities. Chronic migraine imposes a considerable negative burden that influences psychological health, well-being, activity level, and financial stability of individual family members and entire families and affects a detrimental effect on social and marital relationships. Specifically, Buse et al. [39] found that spouses who have a partner with migraine complain of more frequent quarrels, a more likely tendency not to have children or to postpone the occurrence, and the general perception of themselves as a better partner if they had not suffered from headache. Respondents also reported negative repercussions on their working careers and concerns about their finances [40,41]. Regarding both the private and work spheres, patients with chronic migraine report greater degrees of distress than those with episodic migraine [42]. Therefore, migraine represents an important public health problem. Clinical management of migraine aims to reduce the high impact of disability on patients’ lives. Randomized control trials have confirmed the effectiveness of monoclonal antibody treatment in reducing symptoms and improving quality of life [43]. In particular, erenumab effectively reduces monthly migraine days in episodic and chronic migraine [20]. The main limitation of this study is the lack of a control group or a placebo group for doing multiple comparison adjustments or multidimensionality reduction in order to avoid false discovery rate. Another important limitation is the small number of subjects included in this pilot study that does not permit the generalization of the clinical results. In addition, there is the lack of investigation track changes in other medications taken by the study participants.

Despite numerous limitations, however, the results, although preliminary, can be considered useful for a broader evaluation of CGRP therapy. Indeed, migraine is a multidimensional disorder, and understanding the benefits of CGRP therapy may be an important achievement for the psychophysical well-being of patients. Migraine is a disabling disease mainly affecting young women. The rationale of preventives is to reduce disease-related disability and the impact that the disease has on the quality of life. Migraine and headache cause significant limitations in all activities and in all roles of the individual, with obvious consequences on the emotional, behavioral, and social aspects. Given the complexity of the specific subject, we propose to extend the investigation, trying to increase the sample size and introducing the analysis of variables that would permit a better comprehensive assessment. Further studies with greater methodological refinement could establish whether cognitive and psychological dysfunction are associated with an underlying specific migraine pathophysiological process and the efficacy of erenumab on the improvement of these alterations. There is a significant unmet need for effective therapies that address the current limited preventive therapeutic options, characterized by frequent unsatisfactory responses, side effects, and a consequent poor adherence of patients.

## Figures and Tables

**Table 1 medicina-59-00936-t001:** Sample characteristics at baseline.

ID	Gender	Age	MOCA	HAM-A	BDI	MIDAS	SF-36 (PD)	SF-36 (MD)
**01**	F	46	20	9	11	31	33	41
**02**	F	44	26	12	4	145	29	50
**03**	F	59	17	26	4	180	29	20
**04**	F	49	25	13	5	100	28	41
**05**	F	45	28	2	1	32	22	55
**06**	F	51	24	14	18	270	22	48
**07**	F	45	30	15	16	63	37	40
**08**	F	64	15	38	10	60	51	40
**09**	M	66	24	7	8	20	40	42
**10**	F	69	27	20	11	160	30	19
**11**	F	35	20	16	14	55	42	20
**12**	M	52	21	24	11	47	38	25
**13**	F	50	27	8	3	47	34	48
**14**	F	57	27	7	4	75	15	65

**Legend**: MOCA = Montreal Cognitive Assessment; HAM-A = Hamilton Anxiety Rating Scale; BDI = Beck’s Depression Inventory; MIDAS = Migraine Disability Assessment; SF-36M (PD) = Short Form Health Survey 36 (Physical dimension); SF-36 (MD) = Short Form Health Survey 36 (Mental dimension). Age is in years.

**Table 2 medicina-59-00936-t002:** Statistical comparisons of clinical scores between baseline (T0) and follow-up (T1). Scores are in median (first-third quartile); significant differences are in bold.

ASSESSMENT	BASELINE	FOLLOW-UP	*p*-value	ES
**MOCA**	24.50 (20.25–27.00)	27.00 (22.75–29.00)	**<0.01**	−0.65
**HAM-A**	13.50 (8.25–19.00)	14.00 (10.00–13.00)	0.67	−0.11
**BDI**	10.50 (4.25–13.25)	11.50 (4.00–13.00)	0.57	−0.05
**MIDAS**	61.50 (47.00–133.75)	48.00 (17.75–70.75)	0.06	0.43
**SF-36 (PD)**	31.50 (28.25–37.75)	39.00 (31.25–48.75)	**0.04**	−0.44
**SF-36 (MD)**	41.00 (28.75–48.00)	48.50 (31.25–50.75)	0.16	0.09
**SF-36 (PF)**	57.50 (37.50–83.75)	75.00 (56.25–85.00)	0.28	−0.16
**SF-36 (RLP)**	0.00 (0.00–43.75)	25.00 (0.00–100.00)	0.09	−0.33
**SF-36 (P)**	22.00 (22.00–38.75)	46.50 (22.00–64.00)	**0.03**	−0.50
**SF-36 (GH)**	41.00 (30.50–62.75)	58.00 (50.50–71.50)	**0.03**	−0.49
**SF-36 (EF)**	42.50 (31.25–52.50)	50.00 (46.25–60.00)	**<0.01**	−0.65
**SF-36 (SF)**	50.00 (37.00–50.00)	50.00 (37.00–84.00)	0.21	−0.21
**SF-36 (RLE)**	49.50 (8.25–66.00)	83.00 (0.00–100.00)	0.10	−0.32
**SF-36 (EWB)**	60.00 (39.00–81.00)	63.00 (45.00–75.00)	0.66	0.10

**Legend**: ES = Effect Size; MOCA = Montreal Cognitive Assessment; HAM-A = Hamilton Anxiety Rating Scale; BDI = Beck’s Depression Inventory; MIDAS = Migraine Disability Assessment; SF-36M (PD) = Short Form Health Survey 36 (Physical dimension); SF-36 (MD) = Short Form Health Survey 36 (Mental dimension); SF-36 (PF) = Short Form Health Survey 36 (Physical functioning); SF-36 (RLP) = Short Form Health Survey 36 (Role limitations due to physical health); SF-36 (P) = Short Form Health Survey 36 (Pain); SF-36 (GH) = Short Form Health Survey 36 (General health); SF-36 (EF) = Short Form Health Survey 36 (Energy/fatigue); SF-36 (SF) = Short Form Health Survey 36 (Social functioning); SF-36 (RLE) = Short Form Health Survey 36 (Role limitations due to emotional problems); SF-36 (EWB) = Short Form Health Survey 36 (Emotional well-being).

## Data Availability

The data that support the findings of this study are available on request from the corresponding author.

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
