# Peer review of "Migraine Treatment Using Erenumab: Can Lead to a Cognitive and Psychological Qualitative Improvement?"

_medicina, 2023, doi:10.3390/medicina59050936_

Round 1

Reviewer 1 Report

The investigators report on a very small sample of individuals who received erenumab monthly. The study is referred to as a pilot. It is unclear whether the data derive from routine clinic attendees. That is, was the battery of instruments routinely administered to all patients receiving erenumab, or was this a protocol established with the intent of undertaking a study?

It is stated that ‘we found at most two evaluations for each patient, i.e., at first drug administration (T0) and a fourth drug administration (T1)’ Can the investigators confirm and state that all participants in the study had two assessments (it is not clearly stated as an entry criterion).

I don’t understand what is meant by: ‘ previous studies showed that erenumab has a role in identifying the genesis of migraine pain, vasodilation, and neurogenic inflammation [23]’ . Could the investigators please rewrite this to make the meaning more clear or remove the sentence?

No benefit was noted on mood or anxiety scales. Exclusionary criteria were stated to be absence of ‘a previous diagnosis of depression, or another psychiatric syndrome classified by DSM-5’. I was surprised to see that one subject had a BDI of 40, falling within a range that would be consistent with severe depression. Please comment.

Did the investigators track changes in other medications taken by the study participants. Is so, were any changes observed. If not, does this represent another limitation of the study that warrants a mention?

Are there any additional learnings from the study with respect to the future design of a more definitive study, other than increasing the sample size? 

Author Response

The investigators report on a very small sample of individuals who received erenumab monthly. The study is referred to as a pilot. It is unclear whether the data derive from routine clinic attendees. That is, was the battery of instruments routinely administered to all patients receiving erenumab, or was this a protocol established with the intent of undertaking a study?

yes, the battery of instruments was administered to all outpatients

 It is stated that ‘we found at most two evaluations for each patient, i.e., at first drug administration (T0) and a fourth drug administration (T1)’ Can the investigators confirm and state that all participants in the study had two assessments (it is not clearly stated as an entry criterion).

Yes. all subjects had two evaluations.

I don’t understand what is meant by: ‘ previous studies showed that erenumab has a role in identifying the genesis of migraine pain, vasodilation, and neurogenic inflammation [23]’ . Could the investigators please rewrite this to make the meaning more clear or remove the sentence?

Done

No benefit was noted on mood or anxiety scales. Exclusionary criteria were stated to be absence of ‘a previous diagnosis of depression, or another psychiatric syndrome classified by DSM-5’. I was surprised to see that one subject had a BDI of 40, falling within a range that would be consistent with severe depression. Please comment.

We have changed the score of 40, entered by mistake, to the correct score of 4. As insert in the manuscript “In our opinion, depressive or anxious conditions, once established, take several months to weaken. Unfortunately, the Covid-19 pandemic led to changes in the manage-ment of outpatients suspending routine psychological assessments. It has led to a reduc-tion in visits and the lack of long-term evaluations, involving some limitations of the study, such as the small sample size”.

Did the investigators track changes in other medications taken by the study participants. Is so, were any changes observed. If not, does this represent another limitation of the study that warrants a mention?

No. we have only investigated the effect of erenubam. we added the lack investigation track changes in other medications taken by the study participants among limitation of the study.

Are there any additional learnings from the study with respect to the future design of a more definitive study, other than increasing the sample size? 

yes. we have argued additional learnings in the manuscript

Reviewer 2 Report

This manuscript is easy to read and concerns a pilot study but I have some concerns, namely about the methods and results:

2.3 Statistical analysis: Effect size

-        have authors divided the standardized test statistic by the square root of ‘n’ or the Wilcoxon test statistics?

3. Results

This is a pilot study with few cases; however, a control group should have been used and multiple comparisons adjustments or multidimensionality reduction should have been applied in order to avoid false discovery rate.

Author Response

This manuscript is easy to read and concerns a pilot study but I have some concerns, namely about the methods and results:

2.3 Statistical analysis: Effect size

-        have authors divided the standardized test statistic by the square root of ‘n’ or the Wilcoxon test statistics?

By Wilcoxon test

  1. Results

This is a pilot study with few cases; however, a control group should have been used and multiple comparisons adjustments or multidimensionality reduction should have been applied in order to avoid false discovery rate.

We agree with the reviewer's suggestion but unfortunately we do not have a control group. this aspect was included as the main limitation of the study

Reviewer 3 Report

The objective of this manuscript was to investigate the impact of erenumab on the cognitive and psychological functioning of individuals with migraine. The authors compared clinical and psychometric test scores at baseline and follow-up and discovered a substantial enhancement in cognitive performance and quality of life. The data presented are lucid and insightful, and may prove beneficial to clinical practitioners. However, there are several ways in which the manuscript could be improved.

Firstly, the absence of a placebo group may introduce biases, and this issue should be addressed in the discussion section.

Secondly, while table 1 displays only two SF-36 test data, table 2 encompasses data from all ten SF-36 tests. The reason for this discrepancy should be clarified.

Finally, each test has its own strengths and limitations, and it would be pertinent to discuss how these limitations may have impacted the outcomes of the study.

Author Response

Comments and Suggestions for Authors

The objective of this manuscript was to investigate the impact of erenumab on the cognitive and psychological functioning of individuals with migraine. The authors compared clinical and psychometric test scores at baseline and follow-up and discovered a substantial enhancement in cognitive performance and quality of life. The data presented are lucid and insightful, and may prove beneficial to clinical practitioners. However, there are several ways in which the manuscript could be improved.

Firstly, the absence of a placebo group may introduce biases, and this issue should be addressed in the discussion section.

the absence of a grontrol groups have describted as a principal limitations of the study

Secondly, while table 1 displays only two SF-36 test data, table 2 encompasses data from all ten SF-36 tests. The reason for this discrepancy should be clarified.

table 1 shows the total scores of two ppricipal dimension (SF-36M (PD) = Short Form Health Survey 36 (Physical dimension); SF-36 (MD) = Short Form Health Survey 36 (Mental dimension). In tablee 2 we report the total scores of each subitem of the test.

Finally, each test has its own strengths and limitations, and it would be pertinent to discuss how these limitations may have impacted the outcomes of the study.

Done

Round 2

Reviewer 1 Report

I am happy with the responses to my questions/concerns. The new material at the end of the discussion needs to be edited for comprehensibility and correct English.